MINIREVIEW
Applied and Environmental Science

# 2019 Novel Coronavirus (COVID-19) Pandemic: Built Environment Considerations To Reduce Transmission

Leslie Dietz,[a] Patrick F. Horve,[a] David A. Coil,[b] Mark Fretz,[a,c] Jonathan A. Eisen,[d,e,f] Kevin Van Den Wymelenberg[a,c]

aBiology and the Built Environment Center, University of Oregon, Eugene, Oregon, USA
bGenome Center, University of California—Davis, Davis, California, USA
cInstitute for Health and the Built Environment, University of Oregon, Portland, Oregon, USA
dDepartment of Evolution and Ecology, University of California—Davis, Davis, California, USA
eDepartment of Medical Microbiology and Immunology, University of California—Davis, Davis, California, USA
fGenome Center, University of California—Davis, Davis, California, USA

Leslie Dietz and Patrick F. Horve contributed equally to this work. Their order in the byline was determined alphabetically by their last name.

**ABSTRACT**  With the rapid spread of severe acute respiratory syndrome coronavirus 2 (SARS-CoV-2) that results in coronavirus disease 2019 (COVID-19), corporate entities, federal, state, county, and city governments, universities, school districts, places of worship, prisons, health care facilities, assisted living organizations, daycares, homeowners, and other building owners and occupants have an opportunity to reduce the potential for transmission through built environment (BE)-mediated pathways. Over the last decade, substantial research into the presence, abundance, diversity, function, and transmission of microbes in the BE has taken place and revealed common pathogen exchange pathways and mechanisms. In this paper, we synthesize this microbiology of the BE research and the known information about SARS-CoV-2 to provide actionable and achievable guidance to BE decision makers, building operators, and all indoor occupants attempting to minimize infectious disease transmission through environmentally mediated pathways. We believe this information is useful to corporate and public administrators and individuals responsible for building operations and environmental services in their decision-making process about the degree and duration of social-distancing measures during viral epidemics and pandemics.

**KEYWORDS**  COVID-19, SARS-CoV-2, building operations, built environment, novel coronavirus

Increased spread of severe acute respiratory syndrome coronavirus 2 (SARS-CoV-2) causing coronavirus disease 2019 (COVID-19) infections worldwide has brought increased attention and fears surrounding the prevention and control of SAR-CoV-2 from both the scientific community and the general public. While many of the precautions typical for halting the spread of respiratory viruses are being implemented, other less understood transmission pathways should also be considered and addressed to reduce further spread. Environmentally mediated pathways for infection by other pathogens have been a concern in buildings for decades, most notably in hospitals. Substantial research into the presence, abundance, diversity, function, and transmission of microorganisms in the built environment (BE) has taken place in recent years. This work has revealed common pathogen exchange pathways and mechanisms that could lend insights into potential methods to mediate the spread of SARS-CoV-2 through BE-mediated pathways.

Coronaviruses (CoVs) most commonly cause mild illness, but they have occasionally, in recent years, led to major outbreaks of human disease. Typically, mutations that

This article followed an open peer review process. The review history can be read here.

Address correspondence to Patrick F. Horve, pfh@uoregon.edu.

Our review on #SARSCoV2 and #COVID19 in the built environment provides guidance for steps that can be taken in the built environment to potentially slow down spread of SARS-CoV-2 and #flattenthecurve during this global pandemic

*[This article was published on 7 April 2020 but required additional changes, now reflected in the Note Added after Publication on p. 11. The changes to the article were made on 6 May 2020.]*

mSystems®

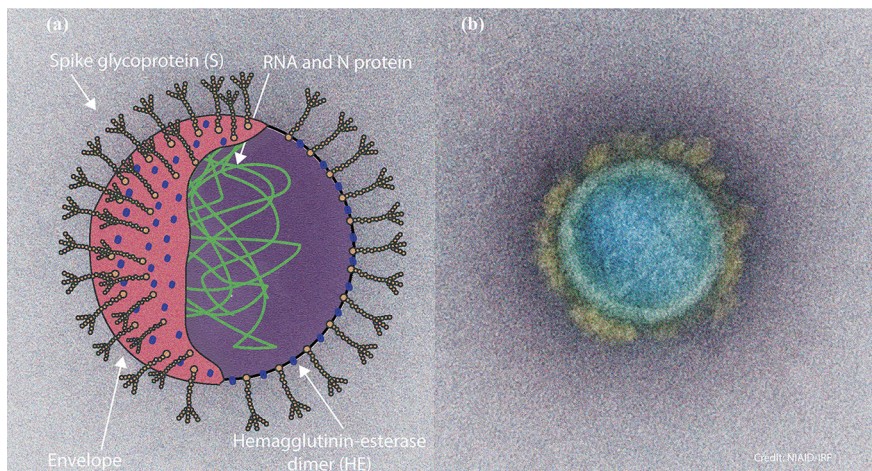

**FIG 1** Structure of SARS-CoV-2 virus. (a) Artistic rendering of the structure and cross section of the SARS-CoV-2 virus (14, 15). (b) Transmission electron micrograph of a SARS-CoV-2 virus particle isolated from a patient and imaged at the NIH, specifically, the National Institute of Allergy and Infectious Diseases (NIAID) Integrated Research Facility (IRF) in Fort Detrick, Maryland (93).

cause structural changes in the coronavirus spike (S) glycoprotein enable binding to new receptor types and permit the jump from an animal host to a human host (1) (zoonotic transmission) and can increase the risk of large-scale outbreaks or epidemics (2). In 2002, a novel CoV, severe acute respiratory virus (SARS), was discovered in the Guangdong Province of China (3). SARS is a zoonotic CoV that originated in bats and resulted in symptoms of persistent fever, chills/rigor, myalgia, malaise, dry cough, headache, and dyspnea in humans (4). SARS had a mortality rate of 10% and was transmitted to 8,000 people during an 8-month outbreak in 2002 to 2003 (5). Approximately 10 years after SARS, another novel, highly pathogenic CoV, known as Middle East respiratory syndrome coronavirus (MERS-CoV), emerged and is also believed to have originated from bats, with camels as the reservoir host (6). MERS-CoV was first characterized in the Arabian Peninsula and spread to 27 countries, having a 35.6% mortality rate in 2,220 cases (7).

**Coronavirus disease 2019 (COVID-19).** In December 2019, SARS-CoV-2, a novel CoV, was identified in the city of Wuhan, Hubei Province, a major transport hub of central China. The earliest COVID-19 cases were linked to a large seafood market in Wuhan, initially suggesting a direct food source transmission pathway (8). Since that time, we have learned that person-to-person transmission is one of the main mechanisms of COVID-19 spread (9). In the months since the identification of the initial cases, COVID-19 has spread to 171 countries and territories, and there are approximately 215,546 confirmed cases (as of 18 March 2020). The modes of transmission have been identified as host-to-human and human-to-human. There is preliminary evidence that environmentally mediated transmission may be possible, specifically, that COVID-19 patients could be acquiring the virus through contact with abiotic BE surfaces (10, 11).

**Epidemiology of SARS-CoV-2.** The *Betacoronavirus* SARS-CoV-2 is a single-stranded positive-sense enveloped RNA virus ( with a genome that is approximately 30 kb in length (12, 13). Spike glycoproteins, the club-like extensions projecting from the cell surface, facilitate the transfer of viral genetic material into a host cell by adhesion (14, 15) (Fig. 1). The viral genetic material is then replicated by the host cell. The infection history of SARS-CoV-2 is believed to have begun in bats with a possible intermediate host of pangolin (16). There are several other betacoronaviruses that occur in bats as a primary reservoir, such as SARS-CoV and MERS-CoV (17). The manifestation of SARS-CoV-2 in a human population occurred late in December 2019, among persons known to frequent a seafood market (18). The first symptoms observed clinically were fever, fatigue, and dry cough, with symptoms ranging from mild to severe (19). Currently, the

protocol developed by the Centers for Disease Control and Prevention (CDC) for diagnosis (20) is a combination of clinical observation of symptoms and a positive result for the presence of the virus using real-time PCR (rt-PCR) (21).

**COVID-19 and the impact of the BE in transmission.** The built environment (BE) is the collection of environments that humans have constructed, including buildings, cars, roads, public transport, and other human-built spaces (22). Since most humans spend >90% of their daily lives inside the BE, it is essential to understand the potential transmission dynamics of COVID-19 within the BE ecosystem and the human behavior, spatial dynamics, and building operational factors that potentially promote and mitigate the spread and transmission of COVID-19. BEs serve as potential transmission vectors for the spread of COVID-19 by inducing close interactions between individuals, by containing fomites (objects or materials that are likely to carry infectious diseases), and through viral exchange and transfer through the air (23, 24). The occupant density in buildings, influenced by building type and program, occupancy schedule, and indoor activity, facilitates the accrual of human-associated microorganisms (22). Higher occupant density and increased indoor activity level typically increase social interaction and connectivity through direct contact between individuals (25) as well as environmentally mediated contact with abiotic surfaces (i.e., fomites). The original cluster of patients were hospitalized in Wuhan, China, with respiratory distress (December 2019), and approximately 10 days later, the same hospital facility was diagnosing patients outside the original cohort with COVID-19. It is presumed that the number of infected patients increased because of transmissions that potentially occurred within the hospital BE (10). The increased exposure risk associated with high occupant density and consistent contact was demonstrated with the COVID-19 outbreak that occurred on the Diamond Princess cruise ship in January 2020 (26). Current estimates of the contagiousness (known as the R0) of SARS-CoV-2, have been estimated from 1.5 to 3 (27, 28). R0 is defined as the average number of people who will contract a disease from one contagious person (29). For reference, measles has a famously high R0 of approximately 12 to 18 (30), and influenza (flu) has an R0 of <2 (31). However, within the confined spaces of the BE, the R0 of SARS-CoV-2 has been estimated to be significantly higher (estimates ranging from 5 to 14), with ~700 of the 3,711 passengers on board the Diamond Princess (~19%) contracting COVID-19 during their 2-week quarantine on the ship (26, 32). These incidents demonstrate the high transmissibility of COVID-19 as a result of confined spaces found within the BE (33). With consideration to the spatial layout of the cruise ship, the proximity of infected passengers to others likely had a major role in the spread of COVID-19 (33).

As individuals move through the BE, there is direct and indirect contact with the surfaces around them. Viral particles can be directly deposited and resuspended due to natural airflow patterns, mechanical airflow patterns, or other sources of turbulence in the indoor environment such as foot fall, walking, and thermal plumes from warm human bodies (22, 34). These resuspended viral particles can then resettle back onto fomites. When an individual makes contact with a surface, there is an exchange of microbial life (35), including a transfer of viruses from the individual to the surface and vice versa (36). Once infected, individuals with COVID-19 shed viral particles before, during, and after developing symptoms (37, 38). These viral particles can then settle onto abiotic objects in the BE and potentially serve as reservoirs for viral transmission (18, 34, 39). Evidence suggests that fomites can potentially be contaminated with SARS-CoV-2 particles from infected individuals through bodily secretions such as saliva and nasal fluid, contact with soiled hands, and the settling of aerosolized viral particles and large droplets spread via talking, sneezing, coughing, and vomiting (34, 40). A study on environmental contamination from MERS-CoV demonstrated that nearly every touchable surface in a hospital housing MERS-CoV patients had been contaminated with the virus (41), and a survey of a hospital room with a quarantined COVID-19 patient demonstrated extensive environmental contamination (18, 34). Knowledge of the transmission dynamics of COVID-19 is currently developing, but based upon studies

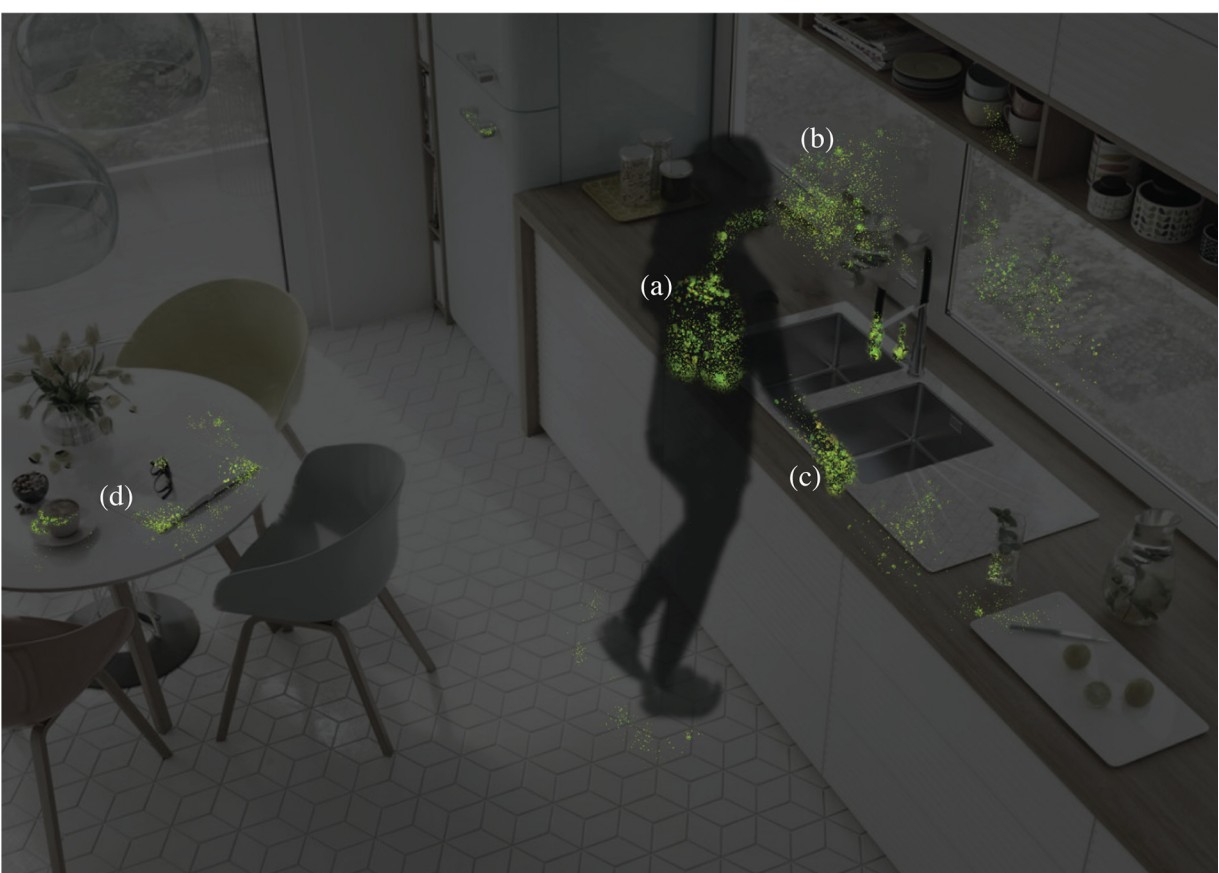

**FIG 2** Conceptualization of SARS-CoV-2 deposition. (a) Once an individual has been infected with SARS-CoV-2, viral particles accumulate in the lungs and upper respiratory tract. (b) Droplets and aerosolized viral particles are expelled from the body through daily activities, such as coughing, sneezing, and talking, and nonroutine events such as vomiting, and can spread to nearby surroundings and individuals (34, 40). (c and d) Viral particles, excreted from the mouth and nose, are often found on the hands (c) and can be spread to commonly touched items (d) such as computers, glasses, faucets, and countertops. There are currently no confirmed cases of fomite-to-human transmission, but viral particles have been found on abiotic BE (built environment) surfaces (34, 39, 42).

of SARS and MERS-CoV, preliminary data on SARS-CoV-2, and CDC recommendations, it seems likely that SARS-CoV-2 can potentially persist on fomites ranging from a couple of hours to 5 days (39, 42, 43) depending on the material (43). Based upon preliminary studies of SARS-CoV-2 survival, the virus survives longest at a relative humidity of 40% on plastic surfaces (half-life median = 15.9 h) and shortest in aerosol form (half-life median = 2.74 h) (43); however, survival in aerosol was determined at a relative humidity of 65%. Based on data related to SARS and MERS, we predict that the viability of SARS-CoV-2 in aerosol is likely longer at lower relative humidity levels. Survival of SARS-CoV-2 at 40% relative humidity on copper (half-life median = 3.4 h), cardboard (half-life median = 8.45 h), and steel (half-life median = 13.1 h) collectively fall between survival in the air and on plastic (43). However, it should be noted that there are no documented cases thus far of a COVID-19 infection originating from a fomite. There is preliminary data demonstrating the presence of SARS-CoV-2 in stool, indicating that transmission can potentially occur through the fecal-oral pathway (18, 29, 34, 44). While transmission of COVID-19 has been documented only through respiratory droplet spread and not through deposition on fomites, steps should still be taken to clean and disinfect all potential sources of SARS-CoV-2 under the assumption that active virus may be transmitted by contact with these abiotic surfaces (34, 39). With an abundance of caution, it is important to consider the possibility that the virus is transmitted through aerosols and surfaces (45). For a conceptualization of SARS-CoV-2 deposition, see Fig. 2.

Previously, it has been confirmed that SARS can be, and is most often, transmitted through droplets (46). Considering that SARS-CoV-2 is from a sister clade to the 2002 SARS virus (47) that is known to transmit from person-to-person, the high incidence of observed person-to-person transmission and the rapid spread of COVID-19 throughout the world and communities, it is accepted at this time that SARS-CoV-2 can also be spread through droplets (13, 48). Based upon previous investigation into SARS (49), spread through aerosolization remains a potential secondary transmission method, especially within the BE. Mitigation of viral transmission through BE air delivery systems is most often reliant on inline filtration media. Residential and commercial systems typically require a minimum efficiency reporting value (MERV) of 8, which is rated to capture 70 to 85% of particles ranging from 3.0 to 10.0 $\mu$m, a strategy employed to minimize debris and loss of efficiency impacts to cooling coils and other heating, ventilation, and air conditioning (HVAC) components. Higher MERV ratings are required to filter incoming outside air based on local outdoor particulate levels. Protective environment (PE) rooms in hospitals require the most stringent minimum filtration efficiency (50). A MERV of 7 (MERV-7) or greater is required as a first filter before heating and cooling equipment, and a second high-efficiency particulate air (HEPA) filter is placed downstream of cooling coils and fans. HEPA filters are rated to remove at least 99.97% of particles at 0.3 $\mu$m in size, representing the most penetrating particle size (51). Most residential and commercial buildings utilize MERV-5 to MERV-11, and in critical health care settings, MERV-13 or higher and HEPA filters are used. MERV-13 filters have the potential to remove microbes and other particles ranging from 0.3 to 10.0 $\mu$m. Most viruses, including CoVs, range from 0.004 to 1.0 $\mu$m (52). However, viruses are rarely observed as individual particles, but instead are expelled from the body already combined with water, proteins, salts, and other components as large droplets and aerosols. Thus far, SARS-CoV-2 has been observed in aerosolized particles in a spectrum of sizes, including 0.25 to 0.5 $\mu$m (96), necessitating high efficiency filtration techniques to reduce the transmission potential of pathogens such as SARS-CoV-2. However, it has been found that gaps in the edges of filters in hospitals has been a contributing factor of the failure of filtering systems to eliminate pathogens from the shared air environment (53).

In recent years, the sharing economy has created environments and added new components to how multiple people share the same spaces. It is possible that infectious disease transmission may be impacted by this shift to the sharing economy. Shared workspaces such as cowork environments, rooms in homes, cars, bikes, and other elements of the BE may increase the potential for environmentally mediated pathways of exposure and add complexity to enacting social-distancing measures. For example, in cases where alternate modes of transportation were previously single occupancy vehicles, these trips are now often replaced with rideshare programs or transportation network companies, the potential for exposure may increase.

**Control and mitigation efforts in the BE.** The spread of COVID-19 is a rapidly developing situation, but there are steps that can be taken, inside and outside the BE, to help prevent the spread of disease. On an individual level, proper handwashing is a critical component of controlling the spread of SARS-CoV-2, other coronaviruses, and many respiratory infections (54–56). Individuals should avoid contact and spatial proximity with infected persons and wash hands frequently for at least 20 s with soap and hot water (39). Furthermore, since it is difficult to know who is infected and who is not, the best way to avoid spread in some situations is by avoiding large gatherings of individuals, also known as "social distancing." At this time, the Food and Drug Administration (FDA) does not recommend that asymptomatic individuals wear masks during their everyday lives to preserve masks and materials for individuals who have been infected with COVID-19 and for health care workers and family that will be in consistent contact with individuals infected with COVID-19 (57). Additionally, wearing a mask can give a false sense of security when moving throughout potentially contaminated areas, and the incorrect handling and use of masks can increase transmission (58). However,

as masks become available, and while prioritizing access to masks for health care workers that are in a higher risk environment daily, wearing a mask would be prudent. There is sufficient evidence to suggest that airborne transmission is possible (49) through aerosolized particles beyond six feet and that a mask would aid in preventing infection through this route.

Since the end of January 2020, many countries have issued travel bans to prevent person-to-person contact and particle-based transmission. These mobility restrictions have been confirmed to help contain the spread of COVID-19 (59). Within local communities, a variety of measures can also be taken to prevent further spread (60). As a whole, these measures are known as non-health-care-setting social-distancing measures. These measures include closing high-occupancy areas such as schools and workplaces. These community-level measures act to prevent disease transmission through the same mechanisms as the worldwide travel restrictions by reducing typical person-to-person contact, decreasing the possibility of fomite contamination by those that are shedding viral particles, and decreasing the possibility of airborne, particle transmission between individuals in the same room or close proximity. These decisions are made by individuals with administrative authority over large jurisdictions, communities, or building stock and are weighed in balance with numerous factors, including health risks and social and economic impacts. Furthermore, despite substantial social-distancing and quarantine practices in place, specific building types and space uses are considered critical infrastructure and essential to maintaining communities, such as health care facilities, housing, and groceries. Better understanding of BE mediating variables can be helpful in decision-making about whether to implement social-distancing measures and for what duration, and to individuals responsible for building operations and environmental services related to essential and critical infrastructure during periods of social distancing, and all building types before and after social-distancing measures are enacted.

Within the BE, environmental precautions that can be taken to potentially prevent the spread of SARS-CoV-2 include chemical deactivation of viral particles on surfaces (39). It has been demonstrated that 62 to 71% ethanol is effective at eliminating MERS, SARS (42), and SARS-CoV-2 (34). This ethanol concentration is typical of most alcohol-based hand sanitizers, making properly applied hand sanitizer a valuable tool against the spread of SARS-CoV-2 in the BE. Items should be removed from sink areas to ensure aerosolized water droplets do not carry viral particles onto commonly used items, and countertops around sinks should be cleaned using a 10% bleach solution or an alcohol-based cleaner on a regular basis. Again, it is important to remember that the main and much more common spread mechanism of previous CoVs has been identified as droplets from talking, sneezing, coughing, and vomiting than by the fecal-oral pathway (34, 38, 39). Administrators and building operators should post signage about the effectiveness of handwashing for at least 20 s with soap and hot water, ensure soap dispensers are full, provide access to alcohol-based hand sanitizer, and implement routine surface cleaning protocols to high-touch surfaces where contamination risks are high, such as around sinks and toilets (39). Most importantly, to prevent the transmission of microbes and thus, undesirable pathogens, it is important to exercise proper handwashing hygiene (39, 61).

Enacting enhanced building HVAC operational practices can also reduce the potential for spread of SARS-CoV-2. Viruses are frequently found associated with larger particles (e.g., complexes with water, proteins, salts, etc.) in a range of sizes. Even though some of these particles have been identified in sizes that could potentially penetrate high efficiency filters, ventilation and filtration remain important in reducing the transmission potential of SARS-CoV-2. Proper filter installation and maintenance can help reduce the risk of airborne transmission, but it is important to understand that filters should not be assumed to eliminate airborne transmission risk. Higher outside air fractions and higher air exchange rates in buildings may help to dilute the indoor contaminants, including viral particles, from air that is breathed within the BE. Higher outside air fractions may be achieved by further opening outside air damper positions

on air-handling units, thus exhausting a higher ratio of indoor air and any airborne viral particles present (62). There are some cautions to consider relative to these building operations parameters. First, increasing outside air fractions may come with increased energy consumption. In the short term, this is a worthwhile mitigation technique to support human health, but building operators are urged to revert to normal ratios after the period of risk has passed. Second, not all air-handling systems have the capacity to substantially increase outside air ratios, and those that do may require a more frequent filter maintenance protocol. Third, increasing airflow rates that simply increase the delivery of recirculated indoor air, without increased outside air fraction, could potentially increase the transmission potential. Higher airflow rates could increase resuspension from fomites and increase the potential for contamination throughout the building by distributing indoor air more quickly, at higher velocities and volumes, potentially resuspending more ultrafine particles (62). Additionally, increasing the indoor air circulation rate could increase the human exposure to viable airborne viral particles shed from other building occupants. Administrators and building operators should collaborate to determine whether increased outside air fractions are possible, what limitations or secondary implications must be considered, and determine a plan around managing the outside air fraction and air change rates.

Increasing evidence indicates that humidity can play a role in the survival of membrane-bound viruses, such as SARS-CoV-2 (63–65). Previous research has found that, at typical indoor temperatures, relative humidity (RH) above 40% is detrimental to the survival of many viruses, including CoVs in general (63, 66, 67), and higher indoor RH has been shown to reduce infectious influenza virus in simulated coughs (67). Based upon studies of other viruses, including CoVs, higher RH also decreases airborne dispersal by maintaining larger droplets that contain viral particles, thus causing them to deposit onto room surfaces more quickly (63, 68, 69). Higher humidity likely negatively impacts lipid-enveloped viruses, like CoVs, through interactions with the polar membrane heads that lead to conformational changes of the membrane, causing disruption and inactivation of the virus (70, 71). Furthermore, changes in humidity can impact how susceptible an individual is to infection by viral particles (72) and how far into the respiratory tract viral particles are likely to deposit (68). Decreased RH has been demonstrated to decrease mucociliary clearance of invading pathogens and weakened innate immune response (72–74). However, RH above 80% may begin to promote mold growth, inducing potentially detrimental health effects (75). Although the current ventilation standard adopted by health care and residential care facilities, ASHRAE 170-2017, permits a wider range of RH from 20% to 60%, maintaining a RH between 40% and 60% indoors may help to limit the spread and survival of SARS-CoV-2 within the BE, while minimizing the risk of mold growth and maintaining hydrated and intact mucosal barriers of human occupants (50, 67). Indoor humidification is not common in most HVAC system designs, largely due to equipment cost and maintenance concerns related to the risk of overhumidification increasing the potential of mold growth. While administrators and building operators should consider the costs, merits, and risks of implementing central humidification, especially during new construction or as a retrofit, it may be too time intensive to implement in response to a specific viral outbreak or episode. In addition, increased RH may lead to increased buildup on filters, decreasing airflow. However, in pandemic situations, this practice likely increases the effectiveness of capturing viral particles, and this benefit outweighs the increased filter maintenance required. Therefore, targeted in-room humidification is another option to consider, and this may reduce the likelihood of a maintenance oversight causing overhumidification.

Building ventilation source and distribution path length can affect the composition of indoor microbial communities. Ventilating a building by introducing air directly through the perimeter of buildings into adjacent spaces is a strategy that does not rely on the efficacy of whole-building filtration to prevent network distribution of microorganisms. Delivering outside air directly through the envelope into an adjacent spatial volume has been shown to increase the phylogenetic diversity of indoor bacterial and

fungal communities and create communities that are more similar to outdoor-associated microbes than air delivered through a centralized HVAC system (76). In some buildings, a similar approach can be accomplished through distributed HVAC units, such as packaged terminal air-conditioners (PTAC) frequently found in hotels, motels, senior housing facilities, condominium units, and apartments or through perimeter passive ventilation strategies such as perimeter dampered vents (77, 78). However, for most buildings, the easiest way to deliver outside air directly across the building envelope is to open a window. Window ventilation not only bypasses ductwork but increases outside air fraction and increases total air change rate as well (79). Administrators and building operators should discuss a plan for increasing perimeter, and specifically window, ventilation when outdoor temperatures are adequate for this practice. Care should be taken to avoid exposing occupants to extreme temperature profiles, and caution should be taken where close proximity would promote potential viral transfer from one residence to another (94, 95).

Light is another mitigation strategy for controlling the viability of some infectious agents indoors. Daylight, a ubiquitous and defining element in architecture, has been shown in microcosm studies to shape indoor bacterial communities in household dust to be less human associated than in dark spaces (80). Moreover, daylight in both the UV and visible spectral ranges reduced the viability of bacteria compared to dark controls in these microcosm spaces (80). In a study simulating sunlight on influenza virus aerosols, virus half-life was significantly reduced from 31.6 min in the dark control group to approximately 2.4 min in simulated sunlight (81). In buildings, much of the sunlight spectrum is filtered through architectural window glass, and the resulting transmitted UV is largely absorbed by finishes and not reflected deeper into the space. Therefore, further research is needed to understand the impact of natural light on SARS-CoV-2 indoors; however, in the interim, daylight exists as a free, widely available resource to building occupants with little downside to its use and many documented positive human health benefits (80–83). Administrators and building operators should encourage blinds and shades to be opened when they are not needed to actively manage glare, privacy, or other occupant comfort factors to admit abundant daylight and sunlight.

While daylight's effect on indoor viruses and SARS-CoV-2 is still unexplored, spectrally tuned electric lighting is already implemented as engineering controls for disinfection indoors. UV light in the region of shorter wavelengths (254-nm UV C [UVC]) is particularly germicidal, and fixtures tuned to this part of the light spectrum are effectively employed in clinical settings to inactivate infectious aerosols and can reduce the ability of some viruses to survive (84). It is important to note that most UVC light is eliminated in the atmosphere, while much of the UVA and UVB spectrum is eliminated through building glass layers. Airborne viruses that contain single-stranded RNA (ssRNA) are reduced by 90% with a low dose of UV light, and the UV dose requirement increases for ssRNA viruses found on surfaces (85, 86). A previous study demonstrated that 10 min of UVC light inactivated 99.999% of CoVs tested, SARS-CoV, and MERS-CoV (87). However, UV germicidal irradiation (UVGI) has potential safety concerns if the room occupants are exposed to high-energy light. For this reason, UVGI is safely installed in mechanical ventilation paths or in upper-room applications to indirectly treat air through convective air movement (88, 89). More recently, far-UVC light in the 207- to 222-nm range has been demonstrated to effectively inactivate airborne aerosolized viruses. While preliminary findings from *in vivo* rodent models and *in vitro* three-dimensional (3-D) human skin models appear favorable to not cause damage to human skin and eyes (90, 91), further research must be conducted to verify the margin of safety before implementation. If implemented safely, UVC and UVGI light offers a range of potential disinfectant strategies for buildings and is a common strategy for deep clean practices in health care settings. Implementing targeted UVC and UVGI treatment may be prudent in other space types where individuals that tested positive for COVID-19 were known occupants, but routine treatment may have unintended consequences and should be implemented with appropriate precaution.

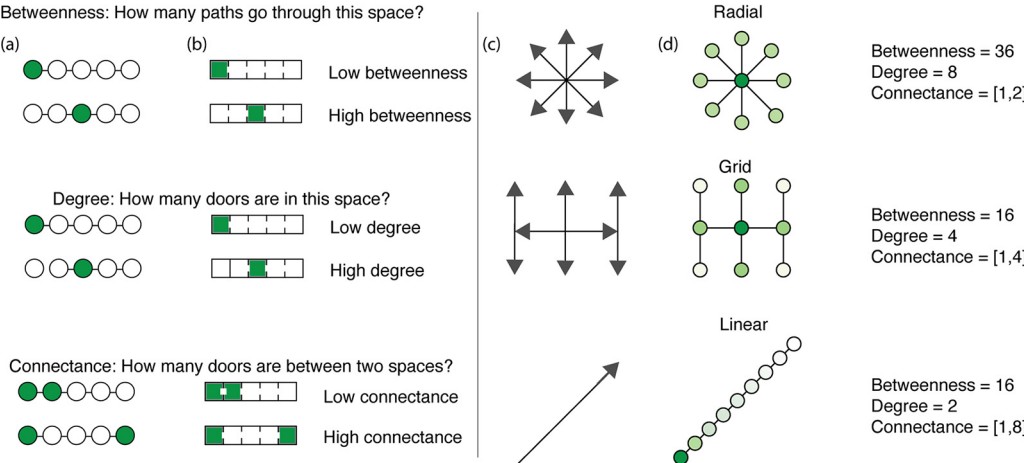

**FIG 3** Spatial connectivity, highlighting betweenness and connectance of common room and door configurations. (a) Circles and lines follow the classic network representation. (b) The rectangles follow the architectural translation of networks. Shaded areas correspond to a measure of betweenness (the number of shortest paths between all pairs of spaces that pass through a given space over the sum of all shortest paths between all pairs of spaces in the building), degree (the number of connections a space has to other spaces between any two spaces), and connectance (the number of doors between any two spaces). (c) The arrows represent possible directions of microbial spread as determined by the layout of the BE. (d) The circles represent the current knowledge of microbial spread based on microbial abundance through BEs as determined by layout. Darker colors represent higher microbial abundance, and lighter colors represent lower microbial abundance.

Spatial configuration of buildings can encourage or discourage social interactions. In recent years, Western society has valued design that emphasizes visual transparency and a feeling of "spaciousness" indoors, whether at home through the use of open plan concepts or at workplaces that harness open office concepts with spatial layouts that intentionally direct occupants to nodes of "chance encounters," thought to enhance collaboration and innovation among employees. While these spatial configurations are culturally important, they may inadvertently enhance opportunities for transmission of viruses through designed human interaction. For example, large, densely populated open office spaces may increase connectivity while private offices may decrease connectivity. Space syntax analysis demonstrates a relationship between spatial disposition and degrees of connectivity (Fig. 3) and has been shown to correlate with the abundance and diversity of microbes within a given space (92). Understanding these spatial concepts could be part of the decision-making process of whether to implement social-distancing measures, to what extent to limit occupant density, and for how long to implement the measures.

**Special considerations for health care settings for current and future epidemics.** Hospitals present unique challenges during the process of mitigating and protecting all inhabitants from an infectious disease outbreak. Not only do health care and hospital facilities have limited options for social-distancing measures to prevent infectious spread, but health care facilities also often cohouse patients with vastly different requirements from the BE around them. For example, high-risk immunocompromised patients are often kept within protective environment (PE) rooms, designed to limit outside airborne infectious agents from entering into the room. To do this, these rooms are positively pressurized, relative to the corridor space, with a minimum of HEPA supply air (ASHRAE 170-2017 [50]). However, this pressurization differential also increases the likelihood that aerosols in the patient room will migrate outside of the PE room and into the higher traffic corridor space when the door is open. While PE rooms typically function as intended for the occupant, if an immunocompromised patient is also under treatment for an airborne infectious disease, the process of limiting pathogen ingress into the room could potentially create involuntary exposure to health care workers, other patients, and visitors via the corridor space. In comparison, airborne infection isolation (AII) rooms utilize a negative pressure differential relative to the

corridor space and adjacent rooms, directly exhausting room air to the exterior of the building to contain aerosolized pathogens from spreading into circulation and shared spaces. The same negative pressure that aids in preventing spread of aerosolized pathogens from inside the room can involuntarily expose the room occupants to airborne pathogens that are sourced from occupants of the corridor space. Both PE and AII rooms may be designed with an anteroom that is used as an additional buffer between common areas and protected spaces to prevent pathogen spread and provide a location for hospital staff to apply and remove personal protection equipment (PPE). However, anterooms are not required for PE or AII rooms and have drawbacks during routine operation; therefore, they exist in only some facilities. They use significant additional floor area, create more travel distance. and increase the visual barrier between patient and rounding care team, therefore, increase costs. These trade-offs might be reconsidered in future design and operational protocols given the high costs of pandemics and the critical role of health care environments during these times.

A discussion of PE and AII rooms does not adequately address the majority of patient rooms within a hospital or health care facility that are not inherently designed with airborne respiratory viruses in mind. Renewed consideration should be given to general facility design to fulfill various requirements for different patient conditions and operational requirements during both routine conditions and disease outbreaks. One such consideration includes separating the means of thermal space conditioning from ventilation provisions. Decoupling these functions permits decentralized mechanical or passive ventilation systems integrated into multifunctional facades with heat recovery and 100% outside air delivery. Mechanically delivering air through the facade would permit all patient rooms to be operated in isolation and individually adjusted to be positively or negatively pressurized, depending on patient requirements, with a higher degree of operational resilience. Furthermore, future designs should reconsider the best way to triage and complete initial assessment of patients that present symptoms related to airborne viruses to minimize exposure to areas with other patient types if possible. In planning for the future, architects, designers, building operators, and health care administrators should aspire for hospital designs that can accommodate periods of enhanced social distancing and minimize connectance and flow between common areas, while also affording flexibility for efficient use of space during normal operating conditions.

**Conclusion.** The number of individuals who have contracted COVID-19 or have been exposed to SARS-CoV-2 has been increasing dramatically. Over a decade of microbiology of the BE research has been reviewed to provide the most up-to-date knowledge into the control and mediation of common pathogen exchange pathways and mechanisms in the BE with as much specificity to SARS-CoV-2 as possible. We hope this information can help to inform the decisions and infection control mechanisms that are implemented by corporate entities, federal, state, county, and city governments, universities, school districts, places of worship, prisons, health care facilities, assisted living organizations, daycares, homeowners, and other building owners and occupants to reduce the potential for transmission through BE mediated pathways. This information is useful to corporate and public administrators and individuals responsible for building design and operation in their decision-making process about the degree and duration of social-distancing measures during viral epidemics and pandemics.

## ACKNOWLEDGMENTS

We thank Jason Stenson, Richard Corsi, Cassandra Moseley, and Linsey Marr for comments on the manuscript. We thank Paul Ward for his graphical contributions.

P.F.H., L.D., and K.V.D.W. conceived of the scope of the article. L.D. and P.F.H. wrote the article, with significant writing contributions from D.A.C. and M.F. P.F.H. developed and created Fig. 1. P.F.H., with outside help, created Fig. 2 and 3. K.V.D.W. and J.A.E. provided significant edits. All authors reviewed the final manuscript.

## Note Added after Publication

After original publication of this paper, several changes were required and have been made in this version of the article.

The text on page 5, first paragraph, line 18, originally read as follows: "HEPA filters are rated to remove at least 99.97% of particles down to 0.3 $\mu$m (51). Most residential and commercial buildings utilize MERV-5 to MERV-11, and in critical health care settings, MERV-12 or higher and HEPA filters are used. MERV-13 filters have the potential to remove microbes and other particles ranging from 0.3 to 10.0 $\mu$m. Most viruses, including CoVs, range from 0.004 to 1.0 $\mu$m, limiting the effectiveness of these filtration techniques against pathogens such as SARS-CoV-2 (52). Furthermore, no filter system is perfect. Recently,. . . ."

The text on page 6, third full paragraph, second sentence, originally read as follows: "Even though viral particles are too small to be contained by even the best HEPA and MERV filters, ventilation precautions can be taken to ensure the minimization of SARS-CoV-2 spread."

The first sentence of Acknowledgments originally said: "We thank Jason Stenson and Cassandra Moseley for comments on the manuscript."

Reference 96 has been added to this version.

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
