## [Reviewer comments · mSystems]

2019 Novel Coronavirus (COVID-19) Outbreak: A Review of the Current Literature and Built Environment Considerations to Reduce Transmission

Leslie Dietz, Patrick Horve, David Coil, Mark Fretz, Jonathan Eisen, and Kevin Van Den Wymelenberg

Corresponding Author(s): Patrick Horve, University of Oregon

Review Timeline:

Submission Date:

March 18, 2020

Accepted:

March 19, 2020

Editor: Jack Gilbert

Transaction Report:

DOI: <https://doi.org/10.1128/mSystems.00245-20>

March 19, 2020

Dr. Patrick Finn Horve
University of Oregon
Biology and the Built Environment Center
Eugene, OR 97403

Re: mSystems00245-20 (2019 Novel Coronavirus (COVID-19) Outbreak: A Review of the Current Literature and Built Environment Considerations to Reduce Transmission)

Dear Dr. Patrick Finn Horve:

Thank you for the timely review - I have read through the whole article and am amazed by the quality of the writing and breadth of the focus. Great work.

Given the relevance to the SARS-CoV-2 outbreak, we are expediting publishing this Review after editorial review, foregoing our normal external review process.

Your manuscript has been accepted, and I am forwarding it to the ASM Journals Department for publication. For your reference, ASM Journals' address is given below. Before it can be scheduled for publication, your manuscript will be checked by the mSystems senior production editor, Ellie Ghatineh, to make sure that all elements meet the technical requirements for publication. She will contact you if anything needs to be revised before copyediting and production can begin. Otherwise, you will be notified when your proofs are ready to be viewed.

Sincerely,

Editor
Editor, mSystems

Journals Department
Phone: 1-202-942-9338